# Deep momentum networks with market trend dynamics

**Jaemin Song, Jaegi Jeon** [ID]*

Graduate School of Data Science, Chonnam National University, Gwangju, Republic of Korea

* jaegijeon@jnu.ac.kr

**Data availability statement:** The raw data underlying this study are from the third-party CHRIS Wiki Continuous Futures dataset, provided by Nasdaq Data Link (https://data.nasdaq.com/data/CHRIS-wiki-continuous-futures/documentation).

## Abstract

Time-series momentum (TSMOM) trading strategies manage positions based on the persistence of return trends. Although long short-term memory (LSTM) deep neural architectures can enhance TSMOM, their performance often deteriorates during abrupt market trend changes. This study aims to improve TSMOM performance, particularly at critical moments marked by significant shifts in long- and short-term trends. To achieve this, we combine short- and long-term signals into a comprehensive market-state representation, employing supervised learning to incorporate these market dynamics into the proposed model. In our experiments, we generate market-state features, referred to as MTDP scores, by numerically capturing changes in market trends via an extreme gradient boosting (XGBoost) process. These MTDP scores are then applied within an LSTM-based trading strategy. A backtest on 99 continuous futures (1995–2021) demonstrates that incorporating MTDP scores enhances the Sharpe ratio, indicating the effectiveness of embedding market-state information in TSMOM. Notably, an 8-week fast momentum look-back window performed best over stable periods (1995–2019). However, during extreme market downturns, such as the COVID-19 crisis, a 20-week fast momentum window not only outperformed shorter windows (4- and 8-week signals) but also recovered more rapidly. These findings suggest that TSMOM strategies can benefit from dynamically adjusting fast momentum windows, consistently generating profitable opportunities even amid turbulent conditions.

## 1 Introduction

Momentum strategies are a critical approach to asset management and investment strategy in financial markets, based on the premise that recent trends in the performance of financial assets often persist. This view contrasts with the traditional efficient market hypothesis [1], which posits that asset prices quickly reflect all available information. Challenging this hypothesis, the study in [2] demonstrated that strategies exploiting past winners (assets with historically high returns) and avoiding past losers (assets with historically low returns) can generate abnormal profits, thereby questioning the efficient market hypothesis and laying the foundation for further momentum-based strategies in the financial sector.

Momentum strategies are typically categorized by how asset portfolios are selected and constructed. One approach, known as cross-sectional momentum, ranks assets by their recent

Access to this dataset is currently restricted by the provider. Researchers interested in obtaining the data should contact Nasdaq Data Link directly for information on their licensing terms and data availability (https://data.nasdaq.com/contact).

**Funding:** This work was supported by the National Research Foundation of Korea (NRF) grant funded by the Korea government (MSIT) (RS-2023-00242528 and RS-2024-00355646). The funders had no role in study design, data collection and analysis, decision to publish, or preparation of the manuscript.

**Competing interests:** The authors have declared that no competing interests exist.

returns and invests accordingly, as explored in [2–4]. In contrast, time-series momentum (TSMOM) bases investment decisions on each asset's own historical performance, a method studied in [5–10].

In [5], extensive backtesting over a 25-year span across equities, currencies, commodities, and bonds revealed strong TSMOM effects, especially concerning persistent return directions and mean reversion over periods exceeding one year. Further, [11] confirmed the persistence of the TSMOM factor and highlighted its cross-market applicability. Meanwhile, [12] underscored the value of volatility scaling for TSMOM strategies, showing outperformance relative to standard buy-and-hold approaches. However, they cautioned that TSMOM in futures markets could suffer when asset-class correlations rise, as evident from a performance dip between 2009 and 2013 following the global financial crisis. These studies demonstrate the potential of TSMOM strategies and the need for continued research and refinement.

In trading, [13] employed a hybrid CNN-LSTM model to predict turning points (TPs) in stock prices, relabeling actions (e.g., buy, sell, hold) via TPs and ordinary points (OPs—moments without a significant trend change), offering a fine-grained approach to market trend analysis. Similarly, in TSMOM contexts, momentum TPs or changepoints have been integrated to enhance strategy design. Building on the deep momentum network (DMN) in [7], the work of [14] incorporated Gaussian process-based changepoints as input features, addressing performance degradation during market shocks and achieving improved Sharpe ratios.

Other studies have linked momentum signals to the detection of market cycles. For instance, [10] developed a dynamic TSMOM strategy that categorizes the market into four phases (bull, correction, bear, rebound) according to the agreement or disagreement between slow (12 months) and fast (one month) momentum signals. By adapting strategy rules to these market cycles, [10] reported Sharpe ratio gains. The study also found that following monthly trends could underperform in correction and rebound phases, aligning with [12]'s observation of post-2008 TSMOM underperformance. Moreover, [10] indicated that periods when slow and fast signals diverge yield predictive insights into future returns, adding a new perspective on TPs.

This paper aims to further improve TSMOM performance by incorporating momentum TPs into a deep neural network framework. Through this integration, the model jointly learns trend estimation and position sizing, generating asset-specific signals for optimal Sharpe ratios. As a benchmark, we adopt the LSTM-based DMN [7], which showed strong results among previously proposed deep neural architectures. Our approach injects predictive return information to extend the existing DMN, enabling trading signals that more closely align with changing market trends. We introduce the concept of momentum TPs similar to [14], applying them as numerical input features that reflect market state transitions.

The principal contributions of this study are as follows. First, building on the TPs described in [10]—focusing on correction and rebound phases—we extend them to eight distinct market trend dynamic points, enhancing the precision of transition detection. This approach offers more granular frequency control of TPs, which in [10] comprised about 35% of the total signals. Second, we refine the numerical representation of these market trend dynamics in DMNs, using an XGBoost-based probability feature engineering pipeline instead of solely relying on simple statistical methods. Third, we compare multiple models using different fast momentum look-back windows. By running five sets of experiments and averaging outcomes, we assess how market trend dynamics affect strategy performance across various market conditions. Furthermore, results from the COVID-19 period are juxtaposed with typical market contexts to highlight the importance of adapting fast momentum signals to match current environments.

This paper is organized as follows. Sect 2 reviews the benchmark model [7] and momentum TPs. Sect 3 details the proposed model and methodology, while Sect 4 describes the experimental setup, including the dataset, baseline comparisons, and backtesting process. Sect 5 presents and critically discusses the full suite of empirical results including backtest performance, ablation studies, and interpretability analyses. Sect 6 distills the main insights and outlines directions for future research.

## 2 Background

### 2.1 Time-series momentum strategies

The basic TSMOM strategy, as presented by [5], calculates the realized return $r_{t,t+1}^{\text{TSMOM}}$ from time $t$ to $t+1$ as follows:

$$r_{t,t+1}^{\text{TSMOM}} = \frac{1}{N_t} \sum_{i=1}^{N_t} X_t^{(i)} \frac{\sigma_{tgt}}{\sigma_t^{(i)}} r_{t,t+1}^{(i)}, \tag{1}$$

where

$$X_t^{(i)} = \text{sign}\left(r_{t-252,t}^{(i)}\right) \text{ and } \sigma_{tgt} = 40\%.$$

In Eq (1), $N_t$ represents the number of assets in the portfolio at time $t$, $X_t^{(i)}$ indicates the position size of the trading signal for each asset $i$, $r_{t,t+1}^{(i)}$ denotes the return of asset $i$ from time $t$ to $t+1$, $\sigma_{tgt}$ is the annualized target volatility, and $\sigma_t^{(i)}$ is the ex-ante annualized volatility estimator for asset $i$, calculated using a 60-day exponentially weighted moving standard deviation.

The TSMOM strategy tracks market trends using historical returns. Under this scheme, if the asset's past 12-month return is positive, the strategy goes long; otherwise, it goes short, and each position is volatility-scaled. Although [5] used 40% as the annual target volatility, we set $\sigma_{tgt}$ to 15% in this study to align with [7].

### 2.2 Time-series momentum strategies with transaction costs

The TSMOM strategy requires periodic portfolio rebalancing to adjust trading signals based on asset volatility or price trend reversals. Although the strategy in Sect 2.1 does not consider transaction costs, these costs must be considered in real-world scenarios, especially in futures-based strategies, where rollover costs can be significant.

Frequent transactions, a characteristic of algorithmic trading strategies, can adversely affect performance due to their associated costs. Therefore, following the approach by [15], we define the daily turnover $O_t^{(i)}$ of the trading signal $X_t^{(i)}$ as follows:

$$O_t^{(i)} = \sigma_{tgt} \left| \frac{X_t^{(i)}}{\sigma_t^{(i)}} - \frac{X_{t-1}^{(i)}}{\sigma_{t-1}^{(i)}} \right|. \tag{2}$$

This turnover is proportional to the daily difference in $X_t^{(i)}/\sigma_t^{(i)}$. The cost-adjusted return formula proposed by [7] is applied to assess the influence of transaction costs on performance:

$$\tilde{r}_{t,t+1}^{\text{TSMOM}} = \frac{\sigma_{tgt}}{N_t} \sum_{i=1}^{N_t} \left( \frac{X_t^{(i)}}{\sigma_t^{(i)}} r_{t,t+1}^{(i)} - C \left| \frac{X_t^{(i)}}{\sigma_t^{(i)}} - \frac{X_{t-1}^{(i)}}{\sigma_{t-1}^{(i)}} \right| \right), \tag{3}$$

where $C$ denotes the transaction cost. Sect 5 explores how different cost levels affect strategy performance during backtesting.

## 2.3 Momentum turning point

In [10], momentum TPs for TSMOM signals were defined using slow (12-month average returns) and fast (monthly returns) components. Building on prior work that highlights the importance of short-horizon signals at weekly and monthly frequencies [8,16], we adopt a weekly cadence for the fast signal to better capture granular market movements. Concretely, we pair a 52-week slow momentum signal with five fast momentum signals—2, 4, 8, 16, and 20 weeks. These intervals follow an approximately exponential spacing to span a range of trading speeds while keeping the specification parsimonious. This fast–slow signal decomposition is consistent with prior work [16], which documents the relevance of weekly TPs in commodity futures captured by short-horizon momentum signals. Throughout our experiments, we set the risk-free rate to zero for practical convenience, thereby simplifying the computation of excess returns.

Weekly momentum signals are expressed as follows:

$$
\text{SLOW}_w^{(i)} = \frac{1}{k_{\text{SLOW}}^{(i)}} \sum_{w'=w-1}^{w-k_{\text{SLOW}}^{(i)}} r_{w'}^{(i)}, \tag{4}
$$

$$
\text{FAST}_w^{(i)} = \frac{1}{k_{\text{FAST}}^{(i)}} \sum_{w'=w-1}^{w-k_{\text{FAST}}^{(i)}} r_{w'}^{(i)}, \tag{5}
$$

where $k_{\text{SLOW}}^{(i)} = 52$ and $k_{\text{FAST}}^{(i)} = 2, 4, 8, 16, 20$ are the respective look-back windows for the slow and fast signals, respectively, and $r_w^{(i)}$ denotes the weekly excess returns of asset $i$.

As noted in [8,10,16], TPs arise when the slow and fast signals diverge, revealing potential shifts in market trends. This divergence may mark a transition from a downtrend to an uptrend, or vice versa. Over-reliance on slower signals risks missing early trend reversals while depending solely on fast signals might interpret short-lived fluctuations as genuine trend changes. Hence, accurately identifying these momentum TPs is crucial for timely and effective trading decisions.

## 2.4 Market state

Following [10], we define four daily market states (bull, bear, correction, rebound) based on the sign of weekly slow/fast signals in Eqs (4) and (5). Specifically, the daily market state $S_t^{(i)}$ at time $t$ is determined as follows:

$$
S_t^{(i)} = \begin{cases} \text{Bull} & \text{if } \text{SLOW}_w^{(i)} \geq 0 \text{ and } \text{FAST}_w^{(i)} \geq 0, \\ \text{Bear} & \text{if } \text{SLOW}_w^{(i)} < 0 \text{ and } \text{FAST}_w^{(i)} < 0, \\ \text{Correction} & \text{if } \text{SLOW}_w^{(i)} \geq 0 \text{ and } \text{FAST}_w^{(i)} < 0, \\ \text{Rebound} & \text{if } \text{SLOW}_w^{(i)} < 0 \text{ and } \text{FAST}_w^{(i)} \geq 0. \end{cases} \tag{6}
$$

Recalling Eqs (4)–(6), the slow and fast signals are computed over different look-back windows, $\left(k_{\text{SLOW}}^{(i)}\right)$ and $\left(k_{\text{FAST}}^{(i)}\right)$, respectively. A bull state arises when both signals are positive, while a bear state arises when both are negative. Correction and rebound states occur when the slow and fast signals are misaligned, indicating potential TPs in the market trend.

## 3 Method

### 3.1 Market trend dynamics

Effectively capturing changes in market states and integrating them into the trading strategy is crucial for improving the performance of the TSMOM strategy. As discussed in Sect 2, TSMOM heavily depends on the momentum TPs and market states. Therefore, we constructed five models with varying fast momentum signals (2, 4, 8, 16, and 20 weeks) and a 52-week slow momentum signal. By varying the fast signal's look-back window, each model captures distinct durations of bull, bear, correction, and rebound phases. Through these scenarios, we aim to pinpoint critical trend shifts and feed this information into our neural network.

To illustrate how fast signal length impacts state identification, we examine the Henry Hub natural gas (NG) futures on the Chicago Mercantile Exchange during the COVID-19 pandemic (2020-01-02 to 2020-12-31). NG is a highly liquid commodity with significant trading volumes and greater volatility than many other asset classes (e.g., equities, bonds). As shown in Fig 1, shorter fast signals (e.g., 2 weeks) closely track short-term swings, capturing the March downturn, the April rally, and subsequent uptrend fluctuations. By contrast, a 20-week fast signal aligns more with longer-term trends and is less sensitive to short-lived price movements.

Although shorter look-back windows may appear suitable for high-frequency environments, they do not necessarily guarantee superior performance. We discuss the performance trade-offs of different fast signals in Sect 5.

To further refine our analysis, we introduce the concept of market trend dynamic points (MTDPs), which mark pivotal changes in the market states. Fig 2 shows a scatterplot of the 52-week average return ($SLOW_w$) against the 2-week average return ($FAST_w$). Each axis divides the data into four quadrants (bull, rebound, bear, and correction), and red "x" markers highlight abrupt shifts near or across these quadrant boundaries. Although MTDPs often cluster near the axes, they can also appear farther out in cases of sudden price shocks.

For brevity, we do not display all comparative charts for the 4-, 8-, 16-, and 20-week cases. Instead, Table 1 summarizes the count of MTDPs and OPs under each configuration. As the fast signal's look-back window increases, fewer MTDPs are identified, mainly because a longer fast window tends to converge with the slow signal. Hence, the choice of fast signal length significantly influences the resolution of detected trend changes.

### 3.2 Feature engineering for MTDP scores

In this subsection, we describe how to encode MTDPs as numeric features for the neural network. We first divide market trends into eight distinct transition categories and one OP category for stable conditions. Table 2 provides the details: for instance, "BuToRe" indicates a transition from bull to rebound, while "BeToCo" indicates a shift from bear to correction. We exclude direct transitions (e.g., bull-to-bear) because they involve simultaneous changes in both slow and fast momentum signals and are rarely observed.

Next, we use XGBoost [17], a high-performing machine learning technique for multiclass classification, to predict the probability of each MTDP class. In other words, XGBoost outputs a 9-dimensional probability vector, one for each category (OP to ReToBu). We refer to this probability vector as the "MTDP score." These scores then serve as input features to the neural network model.

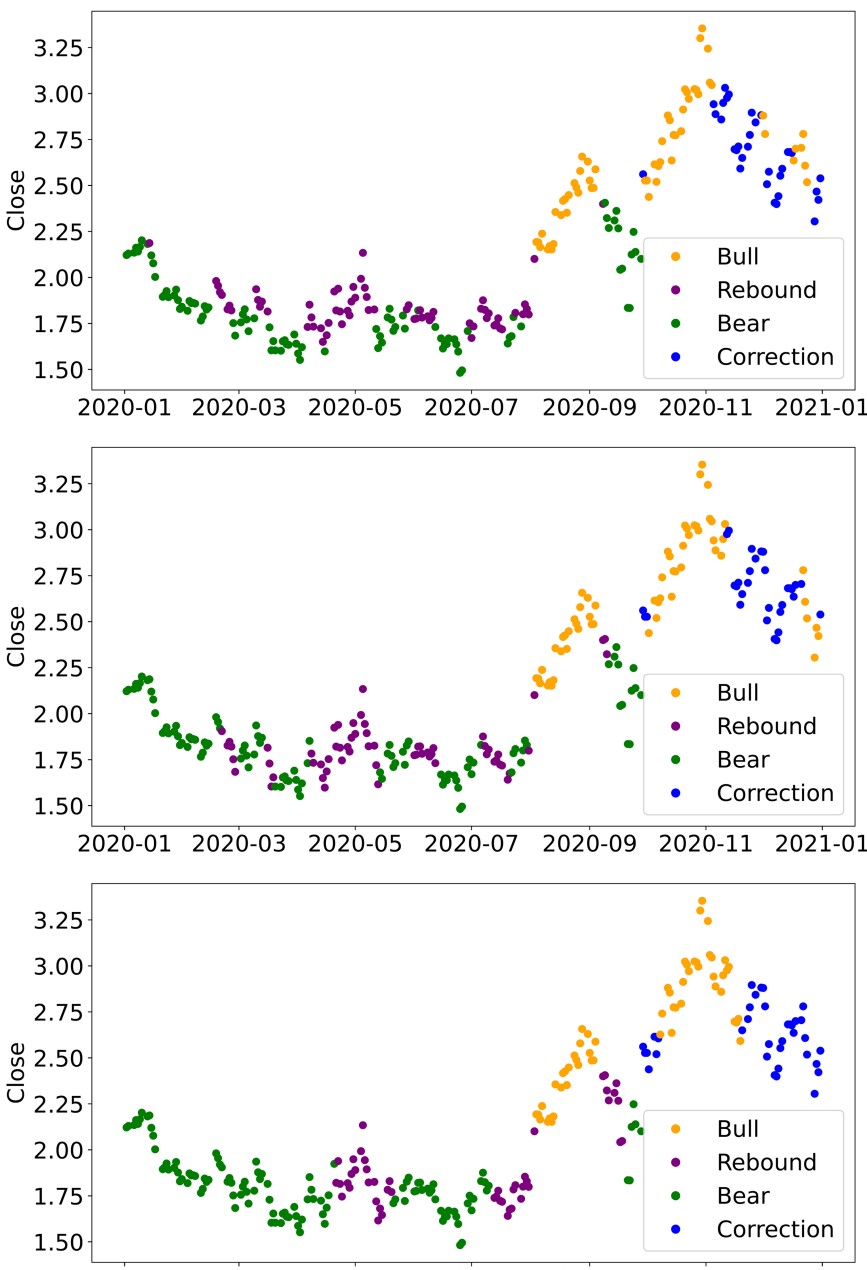

**Fig 1. Market states of Henry Hub natural gas futures during the COVID-19 pandemic.** All panels use a common slow look-back window of $k_{SLOW}$ = 52 weeks; the fast window $k_{FAST}$ varies across panels: 2 weeks (upper), 8 weeks (middle), and 20 weeks (lower). Yellow, purple, green, and blue denote bull, rebound, bear, and correction markets.

The dataset for each asset is split into training (90%), validation (5%), and test (5%) sets. We perform a grid search over key hyperparameters (max depth, gamma, subsample) to minimize the log-loss on the validation set. Early stopping is not used. Furthermore, no random seed was manually fixed, as performance variation across different seeds was negligible. The

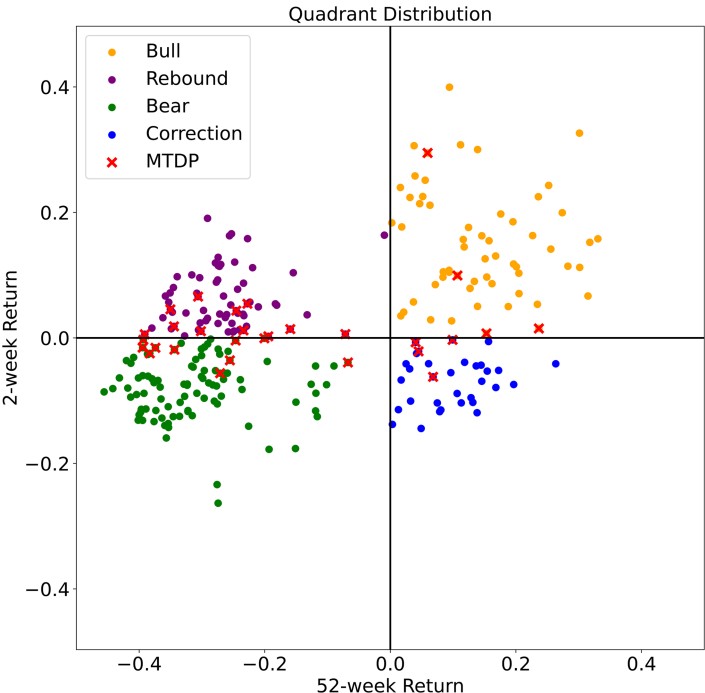

**Fig 2. Scatterplot of the 52-week average return ($SLOW_w$) and 2-week average return ($FAST_w$) for NG during COVID-19.** Four quadrants represent bull (Q1), rebound (Q2), bear (Q3), and correction (Q4) states. The "x" markers (MTDPs) indicate significant trend shifts.

**Table 1. Count of market trend dynamic points (MTDPs) and ordinary points (OPs) for each model of the natural gas futures during the COVID-19 period. Each row reports the number of MTDPs and OPs obtained under the indicated fast–slow momentum look-back windows.**

| Look-back window | | Event counts | |
|---|---|---|---|
| **Fast** | **Slow** | **MTDPs** | **OPs** |
| 2 weeks | 52 weeks | 29 | 224 |
| 4 weeks | 52 weeks | 23 | 230 |
| 8 weeks | 52 weeks | 19 | 234 |
| 16 weeks | 52 weeks | 11 | 242 |
| 20 weeks | 52 weeks | 9 | 244 |

**Table 2. Classification of Market Trend Dynamic Points (MTDPs). Each label represents a specific regime transition and is used to classify MTDPs.**

| Abbreviation | Market Regime Transition |
|---|---|
| OP | Ordinary point (non-transition) |
| BuToRe | Bull → Rebound |
| ReToBe | Rebound → Bear |
| BeToCo | Bear → Correction |
| CoToBu | Correction → Bull |
| BuToCo | Bull → Correction |
| CoToBe | Correction → Bear |
| BeToRe | Bear → Rebound |
| ReToBu | Rebound → Bull |

hyperparameter search space is summarized in Table 3, with asset-wise optimal values listed in S1 Table.

Once training is complete, the XGBoost classifier produces nine probabilities for each data point, indicating how likely it is to belong to each MTDP class. Fig 3 illustrates the entire feature engineering workflow, from labeling the MTDPs to generating the probability-based input features for the neural network.

## 3.3 Deep momentum networks

According to the method proposed by [7], we replace the baseline TSMOM signal $X_t^{(i)}$ in Eq (1) with a direct output from a DMN. This approach integrates position sizing and trend estimation into a single framework. Specifically, we define:

$$r_{t,t+1}^{\text{TSMOM}} = \frac{1}{N_t} \sum_{i=1}^{N_t} X_t^{(i)} \frac{\sigma_{tgt}}{\sigma_t^{(i)}} r_{t,t+1}^{(i)}, \tag{7}$$

**Table 3. Extreme Gradient Boosting (XGBoost) hyper-parameter specification. Predefined settings were fixed, while the remaining three parameters were tuned via grid search on the training set.**

| Category | Parameter | Value/Range |
|---|---|---|
| **Predefined Settings** | Booster type | `gbtree` |
| | Objective | `softprob` |
| | Estimators | 200 |
| | Classes | 9 |
| | Learning rate | 0.02 |
| **Search Range** | Max depth | {3, 6, 9} |
| | Gamma | {0.5, 1.0, 1.5} |
| | Subsample | {0.6, 0.8, 1.0} |

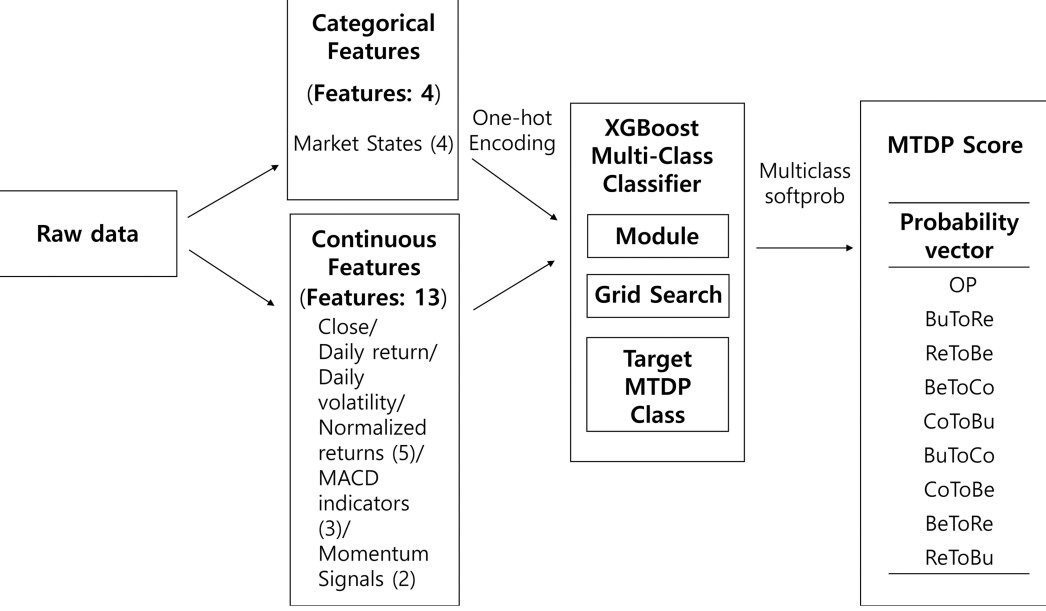

**Fig 3. Overview of the feature engineering process using XGBoost.** The numbers in parentheses next to each feature category denote the count of components or variations used.

$$X_t^{(i)} = f(u_t^{(i)}; \Theta) \text{ and } \sigma_{tgt} = 15\%. \tag{8}$$

Here, $X_t^{(i)}$ is the trading signal at time $t$ generated by the DMN $f$, $u_t^{(i)}$ denotes the input feature vector, $\Theta$ represents network parameters, and $\sigma_{tgt}$ annualized target volatility.

In line with [7], we employ a long short-term memory (LSTM) network [18] for its ability to handle long-range dependencies. As depicted in Fig 4, the model processes 63 sequential input features (Input Feature$_{t-62}$ to Input Feature$_t$) using a single-layer LSTM. The LSTM outputs then pass through a time-distributed dense layer with a tanh($\cdot$) activation, yielding final signals $X_t^{(i)}$ in (−1,1). Each time step's output corresponds to a specific trading position (Position$_{t-61}, \ldots,$ Position$_{t+1}$), thus unifying position sizing with trend learning.

## 4 Experiment

The LSTM architecture includes input, forget, and output gates, along with a cell state, enabling the network to selectively retain or discard information. Let $W$ and $V$ be weight matrices, $\boldsymbol{b}$ be biases, and $\sigma(\cdot)$ and tanh($\cdot$) the sigmoid and hyperbolic tangent activations, respectively. Then:

$$G_{\text{input}}^{(i)}(t) = \sigma(W_i u_t^{(i)} + V_i h_{t-1}^{(i)} + \boldsymbol{b}_i), \tag{9}$$

$$G_{\text{forget}}^{(i)}(t) = \sigma(W_f u_t^{(i)} + V_f h_{t-1}^{(i)} + \boldsymbol{b}_f), \tag{10}$$

$$G_{\text{output}}^{(i)}(t) = \sigma(W_o u_t^{(i)} + V_o h_{t-1}^{(i)} + \boldsymbol{b}_o), \tag{11}$$

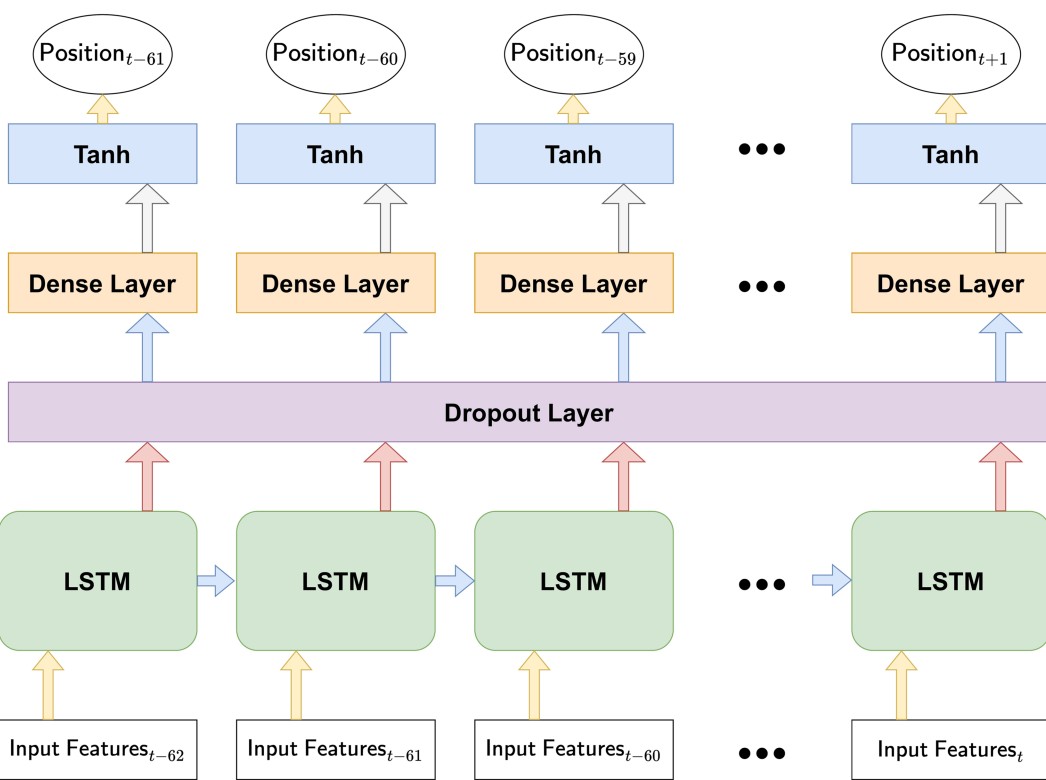

**Fig 4. Flow of deep momentum networks.** Each time step in the LSTM produces a position output ($X_t^{(i)}$), integrating position sizing with trend estimation.

$$\tilde{c}_t^{(i)} = \tanh\left(W_c u_t^{(i)} + V_c h_{t-1}^{(i)} + \boldsymbol{b}_c\right), \tag{12}$$

$$c_t^{(i)} = G_{\text{forget}}^{(i)}(t) \odot c_{t-1}^{(i)} + G_{\text{input}}^{(i)}(t) \odot \tilde{c}_t^{(i)}, \tag{13}$$

$$h_t^{(i)} = G_{\text{output}}^{(i)}(t) \odot \tanh\left(c_t^{(i)}\right), \tag{14}$$

$$X_t^{(i)} = \tanh\left(W_x h_t^{(i)} + \boldsymbol{b}_x\right). \tag{15}$$

Here, $c_t^{(i)}$ is the cell state, $h_t^{(i)}$ is the hidden state, and $\odot$ denotes elementwise multiplication.

To optimize the DMN toward higher cumulative returns, we follow [7] and adopt the Sharpe ratio as the loss function. The Sharpe ratio measures the excess return per unit of volatility, thus balancing expected returns against risk (volatility). However, for simplicity, we set the risk-free rate to zero in this study. Formally, the loss function with zero risk-free rate is defined as:

$$\mathcal{L}_{\text{Sharpe}}(\Omega; \Theta) = -\frac{\sqrt{252}\,\mathrm{E}_\Omega\left[X_t^{(i)} \frac{\sigma_{tgt}}{\sigma_t^{(i)}} r_{t,t+1}^{(i)}\right]}{\sqrt{\mathrm{Var}_\Omega\left[X_t^{(i)} \frac{\sigma_{tgt}}{\sigma_t^{(i)}} r_{t,t+1}^{(i)}\right]}}, \tag{16}$$

where $\Omega$ represents the set of assets at time $t$. By maximizing this Sharpe ratio objective, the DMN learns to generate trading signals that balance return and risk on an ongoing basis.

This section presents the data collection, preprocessing, feature configuration, and backtesting procedures used to evaluate our TSMOM strategy. We first outline the sources of the dataset and preprocessing methods, then describe the feature construction for the DMN. Finally, we explain our structured backtesting scenarios and the hyperparameter tuning approach.

## 4.1 Overview of the dataset

Our experiments use the CHRIS Wiki Continuous Futures dataset covering January 1989 to June 2021 (due to data availability constraints). Sourced from the Nasdaq Data Link, this dataset provides 99 ratio-adjusted continuous futures contracts across four major asset classes (commodities, equity indices, fixed income, and currencies) traded on major global exchanges such as the Chicago Mercantile Exchange, the Intercontinental Exchange, the London International Financial Futures and Options Exchange, and Eurex Exchange. A complete list of these futures, along with their identifiers and brief descriptions, is provided in S2 Table.

Unlike standard short-duration futures, continuous futures overcome contract expiration by rolling over positions, thus creating extended price histories suitable for long-term trend analysis. This characteristic is critical to our study, as it ensures a consistent price series on which to evaluate momentum strategies. We mitigate outliers in daily returns by capping any value that exceeds five times the exponentially weighted moving standard deviation (with a 252-day half-life). After this preprocessing step, our final dataset spans January 1990 to June 2021, providing a comprehensive basis for backtesting the TSMOM strategy.

## 4.2 Feature construction for DMN inputs

We assemble features that combine traditional momentum indicators with the MTDP score, enabling our network to learn the TSMOM strategy better. Building upon the benchmark model of [7] and the methods described in Sect 3.2, we categorize the feature set into three main components:

**Normalized returns:** We compute returns over various periods (daily, monthly, quarterly, semiannual, annual) and normalize each by its corresponding volatility at scale $s \in \{1, 20, 63, 126, 252\}$. This helps ensure a consistent trend representation across different time horizons. Specifically,

$$r_{t-s,t}^{(i)}/(\sigma_s^{(i)}\sqrt{s}),$$

where $r_{t-s,t}^{(i)}$ is the return from $t-s$ to $t$, and $\sigma_s^{(i)}$ is the standard deviation of returns at scale $s$.

**Moving Average Convergence/Divergence (MACD) Indicators:** Following [19], we construct MACD-based signals $M_t^{(i)}(S_k, L_k)$ using short time windows $S_k \in \{8, 16, 32\}$ and long time windows $L_k \in \{24, 48, 96\}$. Specifically, we configured MACD indicators $M_t^{(i)}(8, 24)$, $M_t^{(i)}(16, 48)$, and $M_t^{(i)}(32, 96)$. The MACD is defined by

$$M_t^{(i)}(S_k, L_k) = \frac{q_t^{(i)}}{\text{std}(q_{t-252,t}^{(i)})}, \tag{17}$$

$$q_t^{(i)} = \frac{\text{MACD}(i, t, S_k, L_k)}{\text{std}(p_{t-63,t}^{(i)})}, \tag{18}$$

$$\text{MACD}(i, t, S_k, L_k) = m_t^{(i)}(S_k) - m_t^{(i)}(L_k), \tag{19}$$

where $m_t^{(i)}(S)$ denotes the exponentially weighted moving average (EWMA) of the asset's price at time $t$ with a time scale $S$ corresponding to a half-life of HL $= \log 0.5/\log\left(1 - \frac{1}{S}\right)$. Moreover, $\text{std}(p_{t-63,t}^{(i)})$ represents the 63-day rolling standard deviation of the price, and $q_{t-252,t}^{(i)}$ is the 252-day rolling average of $q_t^{(i)}$. These MACD-based indicators capture medium-range momentum behavior.

**MTDP score:** We define nine classes for market-state transitions and compute the probability of each class using XGBoost (see Sect 3.2). These probabilities form the MTDP score, offering a dynamic view of short- and long-term momentum interplay. For each model variant, we set the fast look-back window to 2, 4, 8, 16, or 20 weeks, paired with a 52-week slow look-back window. The resulting probability vectors serve as additional input features to the deep learning model.

By integrating these normalized returns, MACD indicators, and MTDP scores, our feature set captures both traditional momentum signals and the nuanced dynamics of market-state transitions.

## 4.3 Backtesting setups

We evaluate our strategy against a conventional TSMOM baseline under three backtesting setups (Setup 1–3), summarized below and visualized in Fig 5.

**Setup 1: Long-horizon (1995–2019).** The initial model is trained on 1990–1994, validated on the final 10% of that span, and tested on 1995–1999. The origin then rolls forward by one year; the procedure repeats 20 times, following the guidelines of [20] for mitigating over-fitting in financial backtests (see Fig 5, upper panel).

**Setup 2: COVID-19 stress test (2020).** To isolate the turbulent COVID-19 period, we train/validate on 2015–2019 and evaluate exclusively on calendar year 2020 (Fig 5, middle panel). This setup probes the strategy's behavior amid the rapid regime shifts of the COVID-19 period, which was characterized by frequent market reversals due to misaligned fast and slow momentum signals.

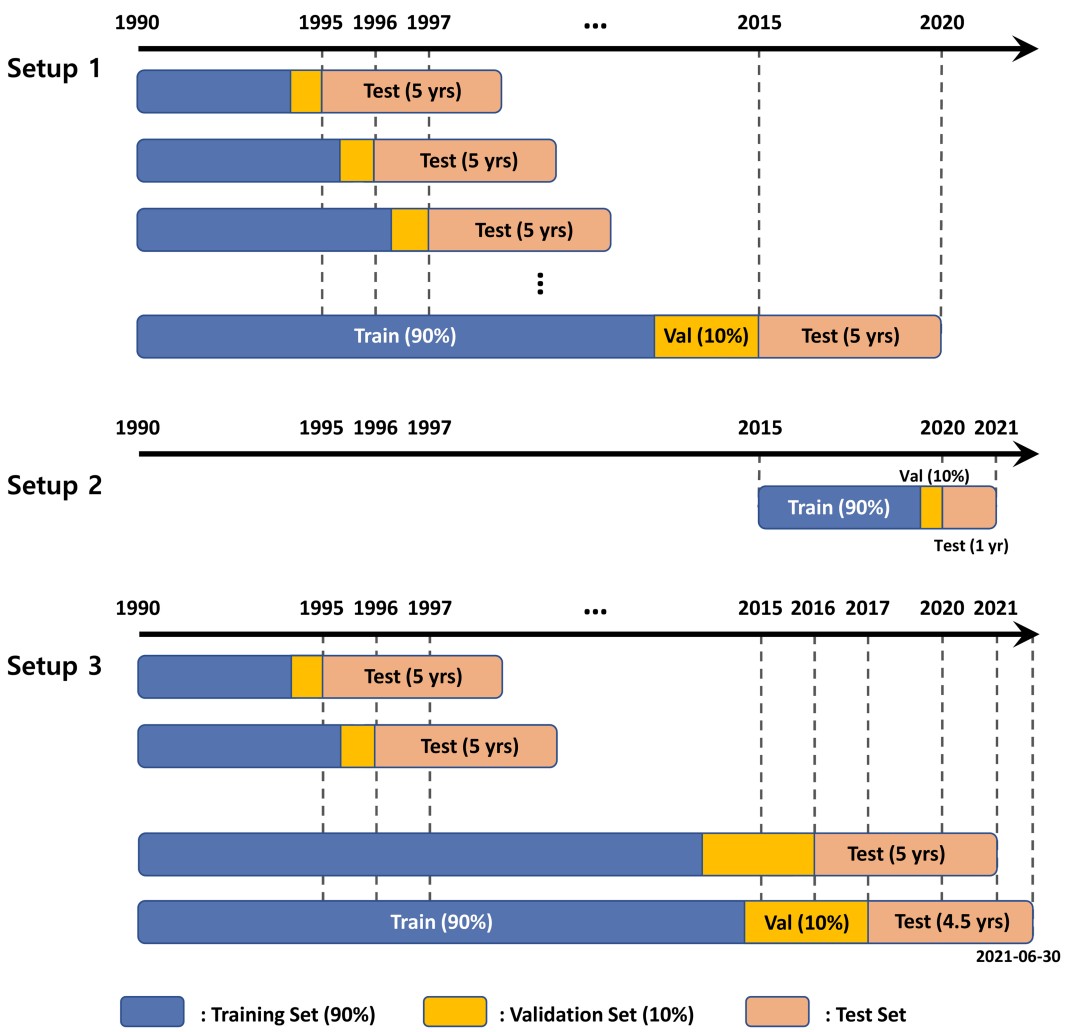

**Fig 5. Backtesting Setups 1–3.** Setup 1 (upper panel) applies a rolling-origin expanding-window split from 1995 to 2019. Setup 2 (middle panel) is a single hold-out stress test on calendar-year 2020, capturing COVID-19 turbulence. Setup 3 (lower panel) extends Setup 1 through 30 June 2021 while retaining the same expanding-window protocol.

**Setup 3: Long-horizon + pandemic (1995–2021-06).** Setup 1 is extended through 30 June 2021, thereby combining a multi-decade perspective with the pandemic aftermath (Fig 5, lower panel). The same rolling-origin expanding-window protocol with a fixed 5-year test horizon is retained.

Fig 5 visualizes the rolling-origin expanding-window protocol used in Setup 1 and 3:

- **Fixed test window:** the out-of-sample horizon is typically five years (salmon blocks).
- **Expanding in-sample window:** the in-sample span grows by one year at each iteration.
- **Train/Validation split:** within the in-sample data we reserve the chronologically last 10% for validation and use the preceding 90% for training; no shuffling is applied.
- **Independent retraining:** for every iteration (and for each setup), model weights are re-initialized and a fresh hyper-parameter search is conducted using only the current

train/validation slice. Hence neither tuned values nor random seeds carry over between runs, precluding information leakage.

Together, the three setups span calm, crisis, and post-crisis conditions, offering a balanced assessment of long-term robustness and short-term adaptability.

### 4.4 Training details

Within each iteration of a setup, the chronologically ordered in-sample data are split 90 : 10 into training and validation subsets (no shuffling). The subsequent five-year block forms the test set, exactly as depicted in Fig 5. Models are trained for up to 300 epochs with early stopping after 25 epochs of no improvement in validation loss.

We optimize the Sharpe-ratio loss defined in Eq (16) using the Adam optimizer [21]. Overfitting is mitigated by dropout [22], whose rate is treated as a searchable hyperparameter. A random search of 100 trials explores dropout rate, hidden-layer width, mini-batch size, learning rate, and maximum gradient norm. The search is conducted on the train + validation subset only (time-ordered hold-out), thereby preserving the strict chronology enforced by the rolling-origin protocol. The explored ranges are summarized in Table 4.

## 5 Results and discussion

### 5.1 Metrics

To evaluate the effectiveness of the trading strategy, we adopt the performance metrics introduced in [7], to analyze portfolio outcomes from 1995 to 2021. These metrics focus on three key aspects:

- **Profitability:** Measured using expected returns and the percentage of positive returns. These indicators capture the average gain and the consistency of profitable outcomes, respectively.
- **Risk:** Evaluated using annualized volatility, downside deviation, and maximum drawdown. Downside deviation measures volatility considering only negative returns, thus indicating downside risk. Maximum drawdown represents the largest loss observed from a peak to a trough in portfolio value.
- **Performance ratios:** Includes the Sharpe, Sortino, and Calmar ratios, as well as the average profit-over-loss ratio. The Sharpe ratio captures excess return per unit of total risk (volatility). The Sortino ratio emphasizes downside risk by considering only negative deviations. The Calmar ratio relates expected return to the maximum drawdown. The average profit-over-loss ratio evaluates the magnitude of gains relative to losses, providing an additional dimension of performance robustness.

**Table 4. Hyper-parameter search ranges for the deep momentum network. Values were explored by a 100-trial random search on the time-ordered train/validation split (90:10); the five-year test window remained unseen during tuning.**

| Hyperparameter | Range |
|---|---|
| Dropout rate | $\{0.1,\ 0.2,\ 0.3,\ 0.4,\ 0.5\}$ |
| Hidden layer size | $\{5,\ 10,\ 20,\ 40,\ 80,\ 160\}$ |
| Minibatch size | $\{64,\ 128,\ 256\}$ |
| Learning rate | $\{10^{-4},\ 10^{-3},\ 10^{-2},\ 10^{-1}\}$ |
| Max gradient norm | $\{10^{-2},\ 10^{0},\ 10^{2}\}$ |

This analysis covers the period from 1995 to 2021. The portfolio performance of each model, adjusted to a target annualized volatility of 15%, is provided in Table 5. In addition, the evolution of the annual Sharpe ratio for each model is listed, allowing a direct comparison of annual performance in Fig 6. Fig 7 explores and visualizes the influence of transaction costs on the overall strategy performance.

Table 5 summarizes the portfolio performance for each model across different scenarios, with a target annualized volatility of 15%. By contrast, Figs 6 and 7, which appear later in this section, will illustrate the annual Sharpe ratio trends and examine the effect of transaction costs, respectively.

## 5.2 Performance evaluation

The proposed model predicts the trading signal $X_t^{(i)}$ of each asset in Eq (8), incorporating the MTDP score (Sects 3.2 and 4.2) into the DMN input. In this subsection, we present the backtesting results for the three setups (Sect 4.3), comparing our model (using MTDP scores) to the benchmark model established by [7]. We consider five model variants, each defined by a different fast look-back window (2, 4, 8, 16, and 20 weeks) and a fixed 52-week slow signal. For simplicity, the benchmark model is denoted "LSTM" and each model is identified by its window length, for example, "the 2-week model."

To verify that the proposed models significantly outperform the benchmark, we conduct one-sided stationary-bootstrap tests [24] with $B = 50{,}000$ resamples. For each asset, the automatic plug-in rule of [23] selects block probabilities clustering around $p \approx 0.50$ ($\approx$ 2-week mean block), consistent with the weak autocorrelation in weekly returns. Asterisks in Table 5

**Table 5. Mean performance metrics over five independent runs and one-sided stationary-bootstrap significance by Backtesting Setup.**

| Model | Expected Return | Volatility | Downside Deviation | Maximum Drawdown | Sharpe Ratio | Sortino Ratio | Calmar Ratio | Positive Return (%) | Avg. Profit / Avg. Loss |
|---|---|---|---|---|---|---|---|---|---|
| **Panel A: Setup 1 (1995–2019)** | | | | | | | | | |
| LSTM | 2.47% | 1.83% | 1.26% | 2.55% | 1.351 | 2.053 | 1.260 | **52.5%** | 1.090 |
| 2-week | 2.45% | **1.64%**<sup></sup> | **1.10%**<sup></sup> | 2.30% | 1.489 | 2.280 | 1.355 | 51.7% | 1.107 |
| 4-week | 2.65% | 1.84% | 1.29%* | 2.70% | 1.526* | 2.319* | 1.422* | 52.0% | 1.090 |
| 8-week | **2.71%** | 1.74%* | 1.17%* | **2.27%** | 1.528* | **2.348** | 1.493* | 50.7% | **1.112** |
| 16-week | 2.47% | 1.83% | 1.31% | 2.75% | 1.337 | 2.029 | 1.250 | 51.7% | 1.072 |
| 20-week | 2.05% | 1.74%* | 1.25% | 2.75% | 1.314 | 1.945 | 1.205 | 51.8% | 1.048 |
| **Panel B: Setup 2 (2020 COVID-19)** | | | | | | | | | |
| LSTM | 1.18% | **3.28%** | **2.64%** | **5.15%** | 0.169 | 0.247 | 0.174 | 57.5% | 0.755 |
| 2-week | 1.10% | 3.79% | 3.09% | 6.16% | 0.245 | 0.298 | 0.135 | 57.8% | 0.772 |
| 4-week | 1.98% | 4.20% | 3.40% | 6.56% | 0.501 | 0.625 | 0.336 | 59.5% | 0.755 |
| 8-week | 1.26% | 4.18% | 3.45% | 7.14% | 0.338 | 0.426 | 0.202 | 59.3% | 0.735 |
| 16-week | 1.79% | 4.20% | 3.43% | 6.77% | 0.436 | 0.541 | 0.275 | **60.5%** | 0.713 |
| 20-week | **2.94%** | 3.71% | 2.99% | 5.71% | **0.696*** | **0.868** | **0.454** | 59.6% | **0.781** |
| **Panel C: Setup 3 (1995–2021-06)** | | | | | | | | | |
| LSTM | 2.44% | 1.86% | 1.30% | 3.00% | 1.306 | 1.977 | 1.195 | 54.0% | 1.069 |
| 2-week | 2.47% | **1.78%**<sup></sup> | **1.22%**<sup></sup> | **2.83%** | 1.389* | 2.140 | 1.302 | 54.1% | 1.079 |
| 4-week | 2.43% | 1.79%* | 1.25%* | 2.95% | 1.411** | 2.181* | 1.351* | 54.1% | **1.082** |
| 8-week | **2.52%** | 1.83%* | 1.28% | 2.87%* | 1.377* | 2.107 | 1.305 | 54.1% | 1.076 |
| 16-week | 2.47% | 1.83%* | 1.26%* | 2.88% | 1.374* | 2.104 | 1.338 | **54.3%** | 1.065 |
| 20-week | 2.32% | 1.79%* | 1.25% | 2.94% | 1.370 | 2.100 | 1.307 | 54.2% | 1.068 |

*Note.* Values are means of five independent runs. * and ** denote one-sided stationary-bootstrap p-values < 0.05 and < 0.01, respectively ($B = 50\,000$ resamples; block probability $p$ selected by the automatic plug-in rule of [23]).

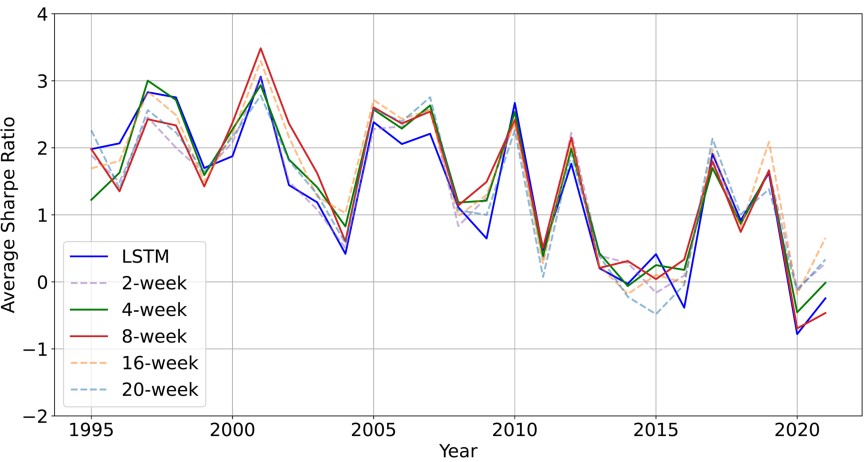

**Fig 6. Comparison of Sharpe ratios between the LSTM benchmark and the proposed models in Setup 3.** The proposed approach generally achieves higher Sharpe ratios than the LSTM baseline.

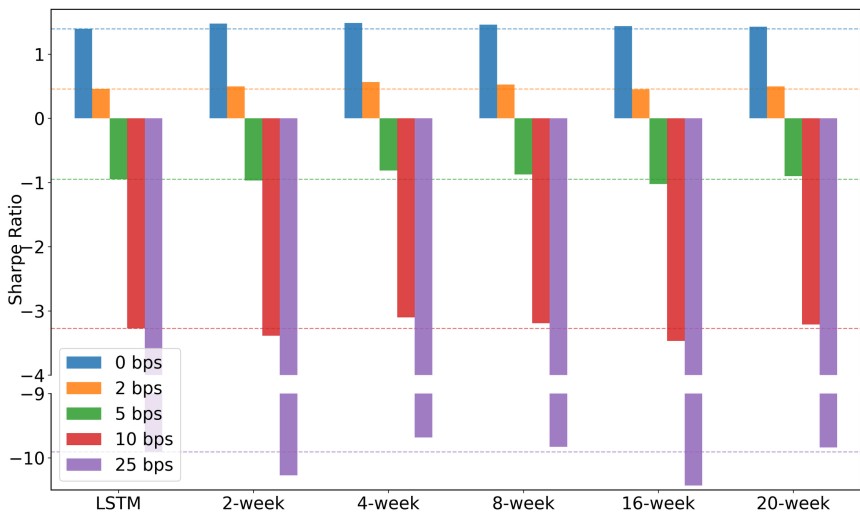

**Fig 7. Impact of transaction costs on the Sharpe ratio.** As transaction costs increase, Sharpe ratios decline for all models; the proposed 4- and 8-week models nonetheless retain a marked edge over the LSTM benchmark.

mark improvements that are significant at the 5% (*) or 1% (**) levels, indicating that the gains are unlikely to be due to random variation.

**Setup 1 (1995–2019).** Using the expanding-window protocol, the model is trained and validated on 1990–1994 data, then tested on the following five-year block; the origin rolls forward 20 times, covering 1995–2019. Panel A of Table 5 reports the mean results over five independent runs.

The 8-week model consistently achieves the best risk-adjusted performance, leading in Sharpe, Sortino, and Calmar ratios. It also records the lowest maximum drawdown and highest expected return. Its Sharpe and Calmar improvements are statistically significant at the 5% level (one-sided stationary bootstrap, 50 000 resamples). The 4-week model also performs well, with significant gains in Sharpe, Sortino, Calmar, and downside deviation. By contrast,

the 2-week model shows slightly lower returns than the benchmark and exhibits the highest volatility. Models with relatively longer fast signals (16 or 20 weeks) yield performances similar to, or marginally below, the benchmark LSTM. This suggests that 4- to 8-week windows capture the market's intermediate trends effectively, balancing short-term noise and long-term drift.

**Setup 2 (2020 COVID-19).** We now focus on the volatile COVID-19 period (2020), using 2015–2019 data for training and validation. As shown in Panel B of Table 5, the 4-week model again surpasses the benchmark in overall performance metrics, similar to the results in Setup 1. Interestingly, 20-week model achieves the highest overall scores and is the only variant whose Sharpe-ratio gain over the LSTM benchmark is statistically significant. No other risk or return metric reaches significance, suggesting that the evidence of outperformance is weaker than in the long-horizon tests. The result nevertheless indicates that during extreme market swings, a longer fast signal (20 weeks) may mitigate whipsaws and preserve capital more effectively.

By contrast, the 2-week model, although reactive, can amplify short-term noise, leading to frequent turning points (MTDPs) and potentially higher turnover or drawdowns. This aligns with the intuition that extremely short windows risk overfitting daily volatility spikes, whereas a mid- or long-term window better withstands abrupt fluctuations.

**Setup 3 (1995–2021-06).** To gauge post-COVID performance in a long-horizon setting, we extend Setup 1's backtesting window 30 June 2021. Panel C of Table 5 indicates that the 4-week model dominates in most risk-adjusted performance measures. Its Sharpe, Sortino, Calmar, downside deviation, and volatility improvements are all statistically significant. Sharpe-ratio gains are also significant for the 2-, 8-, and 16-week models, yet the 4-week model is the only one that attains simultaneous significance in both the Sortino and Calmar ratios, underlining its ability to enhance return while controlling downside risk.

Overall, these findings suggest that no single fast signal length is optimal across all conditions. While the 4-week window often provides a strong balance between responsiveness and stability, the 20-week model may offer greater resilience in highly volatile environments, indicating that adapting or combining signals could further enhance the strategy's robustness.

**Sharpe ratios and transaction costs in setup 3.** Fig 6 plots the average annualized Sharpe ratios for the benchmark and all MTDP models in Setup 3 (1995–2021): the LSTM benchmark appears as a blue solid line, while the best-performing 4- and 8-week MTDP variants are shown as green and red solid lines, respectively. The MTDP-enhanced 4- and 8-week models deliver the most stable risk-adjusted returns: they suffer milder draw-downs after the 2008 crisis and exhibit reduced volatility after 2015. The 16- and 20-week models catch up rapidly during the 2020–2021 recovery, underscoring the value of slower momentum in stress periods. Notably, the bulk of the gain originates in the commodity space: although commodities represent roughly 50% of the investable universe, they contribute more than that proportion to the aggregate Sharpe ratio (see the Sharpe 0 basis points (bps) column of Table 6). Detailed plots of the average annualized Sharpe ratios for all models in Setup 3 are provided separately in S1 Fig.

Having established the cost-free baseline, we now examine how sensitive this outperformance is once realistic trading frictions are imposed. All models are trained and validated under the zero-cost assumption. Once the trading paths are fixed, we apply Eq (3) ex post to the realized daily rebalancings on the test set, deducting round-trip costs of 2, 5, 10, and 25 bps. This post-hoc adjustment leaves the position trajectories, and thus the turnover figures reported for each asset class, exactly as in the 0 bps backtest and measures only the resulting Sharpe loss. The chosen cost grid spans the range typically observed in futures markets, with 25 bps marking the extreme.

Fig 7 illustrates the resulting erosion of performance: Sharpe ratios remain positive up to 2 bps but decline steeply beyond 5 bps, falling below –8 at 25 bps. Table 6 reports the same numbers alongside annual turnover. Turnover is uniformly high ($\approx$ 79–85) because the strategies reallocate full notional weights each day. Fixed Income shows the lowest turnover ($\approx$ 79), while Foreign Exchange reaches the highest ($\approx$ 85), hinting that FX faces the strongest cost drag.

Cost sensitivity differs markedly across asset classes. The MTDP models at 2 bps—especially the 4- and 8-week models—retain a clear edge over the LSTM benchmark. Once costs rise to 5 bps, the 2- and 16-week models start to lag, whereas the 4- and 8-week models still preserve some advantage. At 10 bps and above, every strategy suffers, but the sharpest declines occur in Fixed Income and Foreign Exchange (see the Sharpe 5–25 bps columns of Table 6), confirming that these two sectors are the most vulnerable to transaction-cost erosion. Under the extreme 25 bps scenario the Sharpe ratio of every model turns strongly negative, rendering the strategies economically infeasible.

## 5.3 Feature importance and ablation study on MTDP scores

Building on the performance analysis in Sect 5.2, we investigate which elements of the nine-dimensional MTDP score vector contribute most to the observed performance gains. Specifically, we compute Integrated Gradients (IG) for each score component across Setups 1–3 and retrain a simplified version of the 4-week model in Setup 3 that feeds the model only the Bull and Bear scores,testing whether coarse market state information alone is sufficient for effective position sizing while holding all other settings constant.

Fig 8 reports the average IG-based feature importance for the nine MTDP scores in the 4-week models of Setups 1–3. The IG values are computed by sign-preserving, $L_1$-normalizing, and then averaging them over all runs and rolling windows. The detailed derivation of IG values and full plots for all MTDP models appear in S2 Appendix.

**Table 6. Cost-adjusted Sharpe ratios (0–25 bps) and annualized turnover by asset class.**

| Asset Class | Ann. Turnover | Sharpe 0 bps | Sharpe 2 bps | Sharpe 5 bps | Sharpe 10 bps | Sharpe 25 bps |
|---|---|---|---|---|---|---|
| **Panel A: LSTM** | | | | | | |
| ALL | 83.3 | 1.306 | 0.310 | –1.183 | –3.649 | –10.628 |
| Commodities | 83.1 | 1.346 | 0.886 | 0.195 | –0.954 | –4.377 |
| Equities | 84.1 | 0.458 | 0.071 | –0.508 | –1.469 | –4.287 |
| Fixed Income | 82.4 | 0.366 | –0.723 | –2.318 | –4.287 | –10.189 |
| Foreign Exchange | 84.3 | 0.251 | –0.533 | –1.707 | –3.648 | –9.193 |
| **Panel B: 4-week** | | | | | | |
| ALL | 79.9 | 1.411 | 0.425 | –1.052 | –3.495 | –10.444 |
| Commodities | 79.8 | 1.457 | 0.992 | 0.294 | –0.868 | –4.330 |
| Equities | 80.6 | 0.529 | 0.145 | –0.431 | –1.387 | –4.188 |
| Fixed Income | 78.9 | 0.406 | –0.650 | –2.201 | –4.596 | –10.021 |
| Foreign Exchange | 80.9 | 0.220 | –0.573 | –1.760 | –3.723 | –9.321 |
| **Panel C: 8-week** | | | | | | |
| ALL | 83.8 | 1.377 | 0.381 | –1.110 | –3.577 | –10.586 |
| Commodities | 83.8 | 1.435 | 0.960 | 0.247 | –0.940 | –4.372 |
| Equities | 83.4 | 0.527 | 0.141 | –0.437 | –1.398 | –4.212 |
| Fixed Income | 82.3 | 0.387 | –0.660 | –2.195 | –4.564 | –9.973 |
| Foreign Exchange | 85.3 | 0.190 | –0.616 | –1.823 | –3.816 | –9.492 |

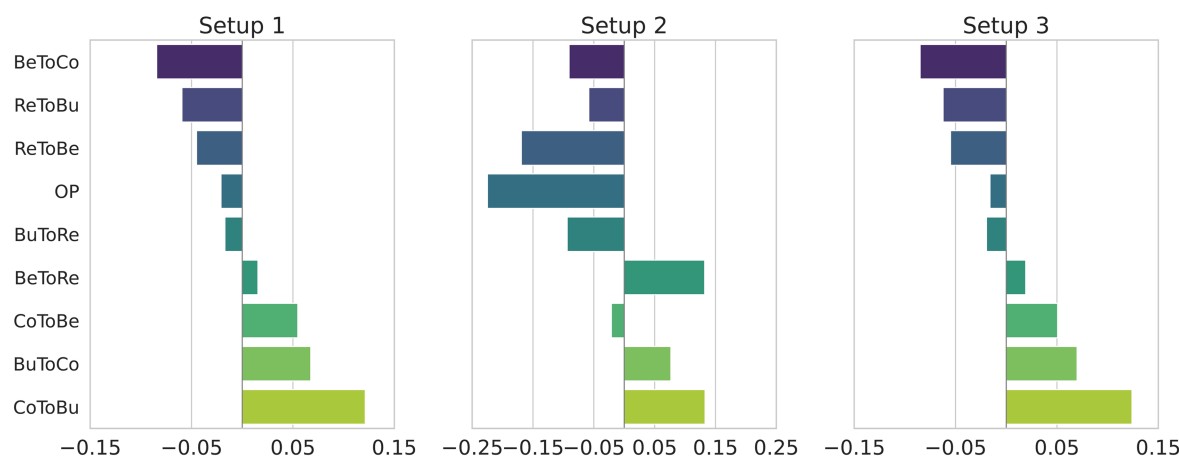

**Fig 8. Average IG-based importance of each of the nine MTDP score components for the 4-week model across Setups 1–3.**

The results in S2 Appendix reveal two key observations:

- In Setups 1 and 3, transitions between Correction and Bull, such as BuToCo (Bull → Correction) and CoToBu (Correction → Bull), tend to be ranked highest. This suggests that probability flows between the Correction and Bull states may offer some of the most influential signals for position sizing.
- In Setup 2, CoToBu scores appear somewhat higher on average, yet no component seems to show consistently dominant importance.

In short, the analysis of long-term data (Setups 1 and 3) suggests that probability flows between the Correction and Bull states are the primary performance drivers. In contrast, it was difficult to identify such distinct patterns in the short-term crisis period (Setup 2).

Next, to gauge what is lost when the model utilizes only simplified Bull–Bear information, we retrain the best-performing configuration (Setup 3 with a 4-week model) after merging the intermediate states (Correction into Bear, Rebound into Bull). This yields a three-state MTDP score whose input channels are:

$$OP \leftarrow \{OP, BeToCo, BuToRe, CoToBe, ReToBu\},$$
$$BeToBu \leftarrow \{BeToRe, CoToBu\},$$
$$BuToBe \leftarrow \{BuToCo, ReToBe\}.$$

The model is then retrained with these three channels while keeping the architecture, optimizer, and evaluation protocol identical to the full nine-state MTDP baseline. As shown in Table 7, the **Ablated** (three-state) model yields slightly higher raw returns than the **Full** (nine-state) model. However, the Full model achieves higher risk-adjusted performance (Sharpe, Sortino, Calmar)—as well as in downside risk measures including volatility and maximum drawdown. While the Ablated model slightly improves raw returns, the Full model consistently outperforms in both risk-adjusted performance and downside stability.

**Table 7. Performance comparison between the MTDP score (Full) and the ablated model using only Bull and Bear transitions (Ablated), evaluated under Setup 3.**

| Model | Expected Return | Volatility | Downside Deviation | Maximum Drawdown | Sharpe Ratio | Sortino Ratio | Calmar Ratio | Positive Return (%) | Avg. Profit / Avg. Loss |
|---|---|---|---|---|---|---|---|---|---|
| Setup 3 (1995–2021-06) | | | | | | | | | |
| Full | 2.32% | **1.45%** | 0.99% | **2.12%** | **1.589** | **2.479** | **1.538** | **54.6%** | 1.087 |
| Ablated | **2.33%** | 1.49% | 0.99% | 2.15% | 1.555 | 2.425 | 1.487 | 54.4% | **1.092** |

*Note.* "Full" uses the complete MTDP score with all nine states; "Ablated" merges Correction into Bear and Rebound into Bull, resulting in a three-state score. The experiments reported in Table 5 were run in 2023 on Tensorflow 2.4 and Keras-tuner 1.0.3, whereas the results above were re-implemented in July 2025 on Tensorflow 2.19.0 and Keras-tuner 1.4.7. Although the same data and random seeds were used, improvements in the software stack lifted the absolute performance levels. The present comparison therefore focuses on each model's relative advantage within the same computational environment.

Taken together with the IG patterns in Fig 8, these results indicate that intermediate-state channels, particularly BuToCo and CoToBu, provide distinctive signals that cannot be faithfully reproduced after collapsing the state space to Bull and Bear. Therefore, the full nine-component MTDP score provides information that is empirically irreplaceable for robust position sizing decisions.

## 6 Conclusion

We refine TPs into a nine–state MTDP framework to capture precise transitions between market states, integrating these MTDP scores as numerical features within a DMN. Our backtests on 99 continuous futures show that this approach is highly effective. Feature importance calculated with IG across all three backtest setups reveals that transitions involving intermediate states are the most influential for position sizing. Furthermore, a Bull–Bear simplification ablation demonstrates that the granular information from the full nine-component MTDP score is not recoverable once the state space is collapsed. Together, the IG evidence and ablation results establish the superiority of the full MTDP specification.

We evaluate five fast look–back windows (2, 4, 8, 16, and 20 weeks) against a fixed 52–week slow window under three backtesting setups. The 4–week model delivers the most consistent performance by balancing responsiveness and noise control. The 2–week model does not guarantee improvement because overly short signals amplify noise, while the 16– and 20–week models yield no uniform gains in ordinary periods. However, the 20–week signal shows resilience during severe drawdowns such as the COVID–19 crisis. These findings indicate that there is no universally optimal fast window and motivate adaptive designs that align the look–back window with prevailing market states.

Because no single window is uniformly optimal, a natural next step is to develop state-conditioned adaptive schemes that re-weight or switch among window-specific experts. Future training should also incorporate cost-aware regularization to target after-cost performance, and practical deployment must enforce regulatory limits such as leverage caps and short-selling restrictions. Promising architectural extensions include Transformer-based models for capturing long-range dependencies and supervised CNNs for more refined trend-change classification, particularly in volatile post-crisis environments.

Finally, while our framework effectively learns dynamics at the individual asset level, it does not account for cross-sectional structures. This limitation may cause it to miss opportunities from synchronized TPs across assets. To address this, a distinct cross-sectional extension could leverage factor models to incorporate common market shocks and co-movements, providing a path toward a more globally aware allocation strategy.

## Supporting information

**S1 Fig.Annualized Sharpe ratios from five experiment iterations.** Each panel displays the results for a different model, allowing for a comparison of their performance consistency. (PDF)

**S1 Appendix. Cross-asset illustrations of MTDP-based position sizing.** Fig A1 shows how MTDP signals guide position sizing and enhance returns for equity and bond futures. Fig A2–A5 extend this to four asset classes across two scenarios, revealing that the best momentum window varies by asset and regime, highlighting the value of adaptive strategies. (PDF)

**S2 Appendix. Interpretability analysis via integrated gradients.** (a) describes the mathematical formulation and normalization process used to compute IG-based feature importances for the MTDP scores. (b) shows the resulting average IG-based feature importances across all models and Setups. (PDF)

**S1 Table. Asset-wise optimal XGBoost hyper-parameters for MTDP models using a 52-week look-back window and fast windows of 2, 4, 8, 16, 20 weeks across 99 assets.** (PDF)

**S2 Table. Asset list and contract descriptions.** (PDF)

## Author contributions

**Conceptualization:** Jaemin Song, Jaegi Jeon.

**Data curation:** Jaemin Song.

**Formal analysis:** Jaemin Song.

**Funding acquisition:** Jaegi Jeon.

**Investigation:** Jaegi Jeon.

**Project administration:** Jaegi Jeon.

**Supervision:** Jaegi Jeon.

**Validation:** Jaegi Jeon.

**Visualization:** Jaemin Song.

**Writing – original draft:** Jaemin Song.

**Writing – review & editing:** Jaegi Jeon.

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
