## [Decision Letter · Decision Letter 0]

13 Jun 2025

PONE-D-25-14477Deep Momentum Networks with Market Trend DynamicsPLOS ONE

Dear Dr. jeon,

Thank you for submitting your manuscript to PLOS ONE. After careful consideration, we feel that it has merit but does not fully meet PLOS ONE’s publication criteria as it currently stands. Therefore, we invite you to submit a revised version of the manuscript that addresses the points raised during the review process. Specifically,

Criterion 3 (Technical quality and detailed description):

The manuscript presents a technically compelling approach, but the absence of publicly available code, lack of detail on final hyperparameter settings, and limited transparency in the feature engineering process hinder reproducibility and independent validation of the results.

Criterion 4 (Conclusions supported by data):

While the conclusions are broadly consistent with the reported findings, they are presented without statistical testing or confidence measures and rely on results from a single asset class, limiting the generalizability of the claims. 

Criterion 7 (Data availability):

The dataset used is not openly accessible, and the reliance on a third-party data provider with restricted access directly conflicts with PLOS ONE’s open data policy, preventing full compliance with community standards for data transparency.

Also, please include discussion on the limitations of the study and provide the next steps. 

We look forward to receiving your revised manuscript.

Kind regards,

Tomo Popovic, Ph.D.

Academic Editor

PLOS ONE

“This work was supported by the National Research Foundation of Korea (NRF) grant funded by the Korea government (MSIT) (RS-2023-00242528 and RS-2024-00355646).”

“This work was supported by the National Research Foundation of Korea (NRF) grant funded by the Korea government (MSIT) (RS-2023-00242528 and RS-2024-00355646).”

“This work was supported by the National Research Foundation of Korea (NRF) grant funded by the Korea government (MSIT) (RS-2023-00242528 and RS-2024-00355646).”

Reviewers' comments:

Reviewer's Responses to Questions

**Comments to the Author**

1. Is the manuscript technically sound, and do the data support the conclusions?

Reviewer #1: Partly

Reviewer #2: Yes

2. Has the statistical analysis been performed appropriately and rigorously? 

Reviewer #1: I Don't Know

Reviewer #2: Yes

3. Have the authors made all data underlying the findings in their manuscript fully available?

Reviewer #1: No

Reviewer #2: Yes

4. Is the manuscript presented in an intelligible fashion and written in standard English?

Reviewer #1: Yes

Reviewer #2: Yes

5. Review Comments to the Author

Reviewer #1: The paper extends the Deep-Momentum Network by injecting nine probabilistic market-state signals— “Market-Trend Dynamic Points” (MTDPs)—and letting an LSTM learn how those states alter momentum exposure. Tests on 99 continuous futures from 1995-2021 show consistent, if moderate, gains in Sharpe, Sortino, and Calmar ratios over the vanilla DMN, indicating that a finer regime lens can sharpen trend-following models.

Several methodological details remain opaque. Hyper-parameter tuning is described as a grid search, yet the specific ranges, random seeds, early stopping rules, and final choices are not reported, so reproducibility and computational cost are hard to judge. The walk-forward protocol in Scenario 3 needs a stricter chronology: wording suggests cross-validation folds may straddle the 2019-2021 boundary, letting pandemic-era information leak into model selection. A purely rolling-origin evaluation would eliminate that risk.

Performance differences would carry more weight with statistical context. Weekly returns are autocorrelated, yet no confidence intervals or bootstrap p-values accompany the Sharpe and drawdown numbers. A stationary bootstrap would clarify whether the 0.05–0.10 uptick in Sharpe is likely to persist. Sensitivity to friction also deserves deeper treatment: Sharpe dips below zero once round-trip costs reach 5 bps for several window settings, whereas real futures trading often exceeds that level. Extending the cost sweep to 2, 5, 10, and 25 bps and reporting turnover by asset class would show where the strategy remains viable.

The nine MTDP probabilities provide a natural hook for attribution, but only qualitative remarks appear. Gradient-based or SHAP analysis highlighting which states drive position shifts, plus an ablation that feeds the LSTM only bull- and bear-state probabilities, would demonstrate the concrete value of the intermediate transitions.

Minor presentation issues persist. Repeated phrases (“…as well as a stronger as well as a stronger momentum effect”), overlapping figure labels, and undefined terms such as “ordinary point (OP)” distract from the core contribution; trimming these and moving dense asset-level plots to an appendix would improve readability.

Reviewer #2: The manuscript presents an innovative and well-executed study on enhancing time-series momentum (TSMOM) strategies using deep momentum networks (DMNs) enriched with market trend dynamic points (MTDPs). The integration of XGBoost-based feature engineering and LSTM networks is methodologically sound and offers valuable insights for both academics and practitioners in algorithmic trading.

1. Clarification on Model Selection Criteria: The choice of five fast momentum look-back windows is reasonable, but further clarification on how these specific intervals were determined would enhance the methodological transparency.

2. Discussion of Model Robustness: The results suggest no universally optimal window size. A brief discussion on potential hybrid or adaptive models that dynamically adjust window lengths could be added in the conclusion or future work section.

3, Figures & Legends: Some figures (e.g., Figures 6 and 7) could benefit from more detailed captions to guide readers unfamiliar with the model architecture.

4. Typos and Formatting: A few minor typographical issues (e.g., “Correaction” instead of “Correction” in Table 2) should be corrected.

6. PLOS authors have the option to publish the peer review history of their article (what does this mean?). If published, this will include your full peer review and any attached files.

Reviewer #1: No

Reviewer #2: No

---

## [Author Response · Author response to Decision Letter 1]

28 Jul 2025

Responses to Reviewer 1

We thank Reviewer 1 for their constructive feedback, which has helped us improve both the clarity and

the scope of the manuscript. We address each comment in turn.

Comment 1. Hyper-parameter tuning is described as a grid search, yet the specific ranges, random seeds, early

stopping rules, and final choices are not reported, so reproducibility and computational cost are hard

to judge.

Response. We have clarified in Section 3.2 (lines 174–179) that early stopping was not employed; model selection

relied solely on validation log-loss. Furthermore, the asset-wise optimal hyperparameters obtained

from the grid search are now provided in the Supporting Information (S1 Table). The hyperparameter

selections for the DMN model, which vary across each fold of the rolling-window backtest, have been

omitted for brevity due to their prohibitive volume. These additions supply the previously missing

details for the XGBoost MTDP classifier.

Comment 2. The walk-forward protocol in Scenario 3 needs a stricter chronology: wording suggests cross-validation

folds may straddle the 2019-2021 boundary, letting pandemic-era information leak into model selection.

A purely rolling-origin evaluation would eliminate that risk.

Response. To eliminate any ambiguity regarding the chronology of Scenario 3, we have substantially revised

“Section 4.3 Backtesting Setups and Section 4.4 Training Details”. Key changes include:

1. The three evaluation schemes are now referred to as backtesting setups, clarifying that they

differ in data-splitting rules.

2. For Setups 1 and 3, we explicitly describe a rolling-origin expanding-window framework: a fixed

5-year test window, an in-sample window that grows annually, and a strictly chronological 90:10

train/validation split (lines 273–277 & 284–299).

3. Setup 2 is now presented as a 2020 hold-out COVID-19 stress test without rolling expansion (lines 278–283).

4. Figure 5 on page 12 has been redrawn to clearly mark fixed 5-year test windows and now includes

a concise summary in the caption.

5. Section 4.4 also clarifies that hyperparameter tuning (100-trial random search) is performed

within each iteration, using only the current train/validation slice; the test set remains untouched throughout (Table 4 caption revised).

These changes ensure that validation folds do not cross the 2019–2021 boundary, and that Setups 1

and 3 adhere strictly to a rolling-origin evaluation.

Comment 3. Performance differences would carry more weight with statistical context. Weekly returns are autocor-

related, yet no confidence intervals or bootstrap p-values accompany the Sharpe and drawdown numbers.

A stationary bootstrap would clarify whether the 0.05-0.10 uptick in Sharpe is likely to persist.

Response. To address the concern regarding statistical inference, we have incorporated a full stationary-bootstrap

analysis and integrated its results into the manuscript. Key additions include:

1. Section 5.2 “Performance Evaluation” (lines 351–356) briefly introduces the one-sided stationary

bootstrap analysis [1], using B = 50,000 resamples and block probability p ≈ 0.50 (mean block

length ≈ 2 weeks) for each asset, selected via the plug-in rule of [2]. We confirm that weekly

returns exhibit weak autocorrelation, justifying the short block length.

2. Table 5 has been retitled and now reports one-sided p-values.

Superscripts * and ** denote significance at the 5% and 1% levels, respectively.

3. Section 5.2 “Performance Evaluation” has been revised to reflect the statistical significance of

Sharpe and related metrics across models and backtesting setups.

These revisions provide the requested statistical context: the observed 0.05–0.10 Sharpe-ratio gains in

Setups 1 and 3 are statistically significant (p < 0.05), while differences in Setup 2 are not.

Comment 4. Sensitivity to friction also deserves deeper treatment: Sharpe dips below zero once round-trip costs

reach 5 bps for several window settings, whereas real futures trading often exceeds that level. Extending

the cost sweep to 2, 5, 10, and 25 bps and reporting turnover by asset class would show where the

strategy remains viable.

Response. We have expanded the manuscript to provide a deeper treatment of transaction-cost sensitivity:

1. “Sharpe Ratios and Transaction Costs in Setup 3” part in Section 5.2 (lines 400–435) now evaluates

round-trip costs of 2, 5, 10, and 25 bps and reports the corresponding cost-adjusted Sharpe ratios

and annual turnover by asset class.

2. Figure 7 visualizes the erosion of Sharpe ratios across the four cost levels, and Table 6 reports

the corresponding cost-adjusted Sharpe ratios together with annual turnover for each asset class.

These additions identify the cost thresholds at which the proposed strategies remain profitable and

pinpoint the asset classes with the greatest exposure to slippage.

Comment 5. The nine MTDP probabilities provide a natural hook for attribution, but only qualitative remarks ap-

pear. Gradient-based or SHAP analysis highlighting which states drive position shifts, plus an ablation

that feeds the LSTM only bull- and bear-state probabilities, would demonstrate the concrete value of

the intermediate transitions.

Response. A new Section 5.3 “Feature Attribution and Ablation Study on MTDP Scores” now provides the

requested quantitative evidence.

1. Sign-preserving, L1-normalised Integrated Gradients (Figure 8; full plots in S2 Appendix) show

that flows between Correction and Bull (BuToCo, CoToBu) are the most influential channels in

the long-horizon setups, directly linking specific MTDP components to position-sizing decisions.

2. Bull–Bear ablation (Table 7) collapsing the nine states to three (Bull, Bear, OP) keeps headline

returns unchanged but lowers Sharpe (1.589 → 1.555), Sortino (2.479 → 2.425), and Calmar (1.583

→ 1.487) ratios, confirming that intermediate transitions contain irreplaceable information.

S2 Appendix documents the IG calculation procedure and presents feature-importance plots for all

MTDP models. The combined gradient analysis and ablation demonstrate that the full nine-component

MTDP score is essential for robust, risk-efficient trading.

Comment 6. Minor presentation issues persist. Repeated phrases (“. . . as well as a stronger as well as a stronger

momentum effect”), overlapping figure labels, and undefined terms such as “ordinary point (OP)” dis-

tract from the core contribution; trimming these and moving dense asset-level plots to an appendix

would improve readability.

Response. Minor issues fixed (duplicates removed, figure labels tidied, OP defined at line 28).

To improve readability, detailed asset-level plots and diagnostics are now in the Supporting Information.

References

[1] Politis DN, Romano JP. The stationary bootstrap. Journal of the American Statistical association. 1994;89(428):1303–1313.

[2] Politis DN, White H. Automatic block-length selection for the dependent bootstrap. Econometric reviews. 2004;23(1):53–70.

Responses to Reviewer 2

We thank Reviewer 2 for their careful reading and valuable comments regarding the motivation, theo-

retical formulation, and contribution of our work.

Comment 1. Clarification on Model Selection Criteria: The choice of five fast momentum look-back windows is

reasonable, but further clarification on how these specific intervals were determined would enhance the

methodological transparency.

Response. We selected the five fast windows (2, 4, 8, 16, and 20 weeks) to follow an approximately exponential

grid, balancing coverage of various time horizons while keeping the model parsimonious.

As now explained in Section 2.3 (lines 99–107), this approach extends prior work to a weekly frequency.

We also clarify that intermediate windows (e.g., 6 and 12 weeks) were excluded for tractability, as

preliminary tests showed they did not materially improve risk-adjusted outcomes while significantly

increasing the hyper-parameter search space.

Comment 2. Discussion of Model Robustness: The results suggest no universally optimal window size.

A brief discussion on potential hybrid or adaptive models that dynamically adjust window lengths could be

added in the conclusion or future work section.

Response. We agree and have expanded the conclusion to outline a regime-aware hybrid design. Specifically,

Section 6 (lines 494–501) now proposes a two-stage architecture that (i) employs a Hidden Semi-

Markov Model to infer latent trend regimes and (ii) routes inputs through a Multi-Task Mixture-

of-Experts network whose expert weights are modulated by the inferred regime probabilities. This

adaptive approach enables smooth re-weighting among 2-, 4-, 8-, 16-, and 20-week experts and will be

the focus of our follow-up study.

Comment 3. Figures & Legends: Some figures (e.g., Figures 6 and 7) could benefit from more detailed captions to

guide readers unfamiliar with the model architecture.

Response. To improve clarity and manuscript flow, we have thoroughly revised our figures and moved the asset-

specific examples to the Supporting Information (S1 Appendix).

– The figures referenced by the reviewer (formerly Figures 6 and 7) are now located in S1 Appendix as Figures A1–A5.

– All captions have been rewritten to be self-contained. For example, Figure A1’s title now specifies

the assets and timeframe ("DMN-based trading strategies for the S&P 500 and Euro-Bobl futures

(2016-–2020)"), while the captions for Figures A2–A5 clarify the exact asset and backtesting setup used in each plot.

This approach ensures each visual is self-explanatory, directly addressing the reviewer’s concern.

Comment 4. Typos and Formatting: A few minor typographical issues (e.g., “Correaction” instead of “Correction”

in Table 2) should be corrected.

Response. We performed a manuscript-wide proofreading pass and fixed all detected issues.

---

## [Decision Letter · Decision Letter 1]

15 Aug 2025

Deep Momentum Networks with Market Trend Dynamics

PONE-D-25-14477R1

Dear Dr. jeon,

We’re pleased to inform you that your manuscript has been judged scientifically suitable for publication and will be formally accepted for publication once it meets all outstanding technical requirements.

Kind regards,

Tomo Popovic, Ph.D.

Academic Editor

PLOS ONE

Additional Editor Comments (optional):

Reviewers' comments:

Reviewer's Responses to Questions

**Comments to the Author**

1. If the authors have adequately addressed your comments raised in a previous round of review and you feel that this manuscript is now acceptable for publication, you may indicate that here to bypass the “Comments to the Author” section, enter your conflict of interest statement in the “Confidential to Editor” section, and submit your "Accept" recommendation.

Reviewer #1: All comments have been addressed

Reviewer #2: All comments have been addressed

2. Is the manuscript technically sound, and do the data support the conclusions?

Reviewer #1: Yes

Reviewer #2: (No Response)

3. Has the statistical analysis been performed appropriately and rigorously? 

Reviewer #1: N/A

Reviewer #2: Yes

4. Have the authors made all data underlying the findings in their manuscript fully available?

Reviewer #1: Yes

Reviewer #2: Yes

5. Is the manuscript presented in an intelligible fashion and written in standard English?

Reviewer #1: Yes

Reviewer #2: Yes

6. Review Comments to the Author

Reviewer #1: All comments have been adressed and all the previous ,,problems" and shortcomings are resolved. The paper is technicaly very sound.

Reviewer #2: (No Response)

7. PLOS authors have the option to publish the peer review history of their article (what does this mean?). If published, this will include your full peer review and any attached files.

Reviewer #1: No

Reviewer #2: No

---

## [Editor Report · Acceptance letter]

PONE-D-25-14477R1

PLOS ONE

Dear Dr. jeon,

I'm pleased to inform you that your manuscript has been deemed suitable for publication in PLOS ONE. Congratulations! Your manuscript is now being handed over to our production team.

Kind regards,

on behalf of

Prof. Tomo Popovic

Academic Editor

PLOS ONE